# Chemokines in the Landscape of Cancer Immunotherapy: How They and Their Receptors Can Be Used to Turn Cold Tumors into Hot Ones?

**DOI:** 10.3390/cancers13246317

**Published:** 2021-12-16

**Authors:** Nathan Karin

**Affiliations:** Department of Immunology, Faculty of Medicine, Technion, P.O. Box 9697, Haifa 31096, Israel; nkarin@technion.ac.il

**Keywords:** chemokines, chemokine receptors, cancer immunotherapy, CXCL9, CXCL10, CCR8, CCL1, regulatory T cells, immune checkpoint inhibitors

## Abstract

**Simple Summary:**

For the last decade, the most successful approach for treating cancer has been the use of monoclonal antibodies to immune checkpoint inhibitors (ICI), also known as immune checkpoint blockers (ICB). Unfortunately, many cancers do not respond to these treatments. “Hot” tumors are those that show signs of inflammation, meaning they have been invaded by effector T cells rushing to fight the cancerous cells. Evidence suggests that the limited success of ICI-based immunotherapies is related to attempts to treat patients with “cold tumors” that either do not contain effector T cells or in which these cells are markedly suppressed by regulatory T cells (T_regs_). Chemokines are a well-defined group of proteins with chemotactic properties. We focus on key chemokines that not only attract leukocytes to tumor sites but also shape their biological properties. We propose using stabilized forms of two of them: CXL9 and CXCL10, to enhance anti-tumor immunity and possibly transform cold tumors into hot ones. Additionally, we discuss the possibility of targeting or deleting a key subset of T_reg_s that are CCR8^+^ T_regs_ and are highly dominant at the tumor site of several cold tumors. This may convert these cold tumors into hot tumors, and thus extend the success of immunotherapy beyond its current limits.

**Abstract:**

Over the last decade, monoclonal antibodies to immune checkpoint inhibitors (ICI), also known as immune checkpoint blockers (ICB), have been the most successful approach for cancer therapy. Starting with mAb to cytotoxic T lymphocyte antigen 4 (CTLA-4) inhibitors in metastatic melanoma and continuing with blockers of the interactions between program cell death 1 (PD-1) and its ligand program cell death ligand 1 (PDL-1) or program cell death ligand 2 (PDL-2), that have been approved for about 20 different indications. Yet for many cancers, ICI shows limited success. Several lines of evidence imply that the limited success in cancer immunotherapy is associated with attempts to treat patients with “cold tumors” that either lack effector T cells, or in which these cells are markedly suppressed by regulatory T cells (T_regs_). Chemokines are a well-defined group of proteins that were so named due to their chemotactic properties. The current review focuses on key chemokines that not only attract leukocytes but also shape their biological properties. CXCR3 is a chemokine receptor with 3 ligands. We suggest using Ig-based fusion proteins of two of them: CXL9 and CXCL10, to enhance anti-tumor immunity and perhaps transform cold tumors into hot tumors. Potential differences between CXCL9 and CXCL10 regarding ICI are discussed. We also discuss the possibility of targeting the function or deleting a key subset of T_regs_ that are CCR8^+^ by monoclonal antibodies to CCR8. These cells are preferentially abundant in several tumors and are likely to be the key drivers in suppressing anti-cancer immune reactivity.

## 1. Introduction

Chemokines are small proteins that have mostly been associated with directing leukocyte migration, and in affecting the dynamics of cancer, inflammation, and immune regulation [1,2,3]. As for cancer, many chemokines are produced by cancer cells that also possess their receptors [4,5]. So far, sixteen out of nineteen human chemokine receptors have been detected in cancer cells [6]. Key examples are CXCR4, CXCR1/2, CCR2, CXCR3, CCR5, and their ligands [1]. All became targets for cancer therapy [1,4,5,7,8,9]. The traditional view has been that chemokines mostly support tumor growth and survival either by a direct effect on tumor cells that possess their receptors [5] or by indirect mechanisms [5,10,11,12,13,14]. These indirect mechanisms mostly include interactions with their receptors on endothelial cells within the tumor microenvironment (TME), to induce growth factors production, and also in attracting bone marrow (BM)-derived cells to the tumor site. These cells then assist tumor growth and suppress the activities of anti-tumor effector T cells that limit tumor growth [5,10,11,12,13,14]. The major BM-derived cells that are known to support tumor growth and suppress anti-tumor immune reactivity are tumor-associated macrophages (TAMS), myeloid-derived suppressor cells (MDSC), neutrophilic cells, and regulatory T cells (T_regs_). All of them suppress anti-tumor immune reactivity, and some of them directly support tumor growth [5,10,11,12,13,14]. Altogether, it implies that chemokines and their receptors are valid targets for cancer therapy [15]. Yet, thus far attempts to block many of these chemokines or their receptors showed limited success in human cancers. A possible mechanism of tumor escape may involve the rapid selection of resistant tumor cells [4]. The other possible explanation could be redundancy between chemokines [16,17].

The breakthrough of using monoclonal antibodies to immune checkpoint inhibitors (ICI) (also referred to as immune checkpoint blockers, ICB) opened new therapeutic opportunities [18,19,20,21,22,23,24,25,26]. The first successful approach of ICI has been the use of anti-cytotoxic T lymphocyte antigen 4 (CTLA-4) inhibitors in metastatic melanoma [25,27,28,29,30], and continuing with blocking the interactions between program cell death 1 (PD-1) and its ligands: program cell death ligand 1 (PDL-1) and program cell death ligand 2 (PDL-2) [25,27,28,29,30]. These blockers have been approved for about 20 different indications [23,26,31,32,33,34,35,36]. As a part of their mechanism of action, these ICIs enhanced the activity of tumor-specific effector CD4^+^ and CD8^+^ T cells [31,32,34,37]. Yet, immune checkpoint therapies (ICT) for many cancer diseases still show limited success [21,31,38,39,40,41]. Moreover, even in diseases with a significant positive response to ICI a relatively high number of patients are poor responders, and/or develop severe immune-related toxicities. This led to intensive research in two complementary avenues. The first focuses on developing tools for personalized-based medicine enabling to predict success on a personalized basis and excludes patients that following therapy have a high risk of developing immune-related toxicities [42,43,44,45,46,47,48]. The other avenue is spending efforts on developing new immunotherapeutic tools that would be used, either alone, or in combination with “conventional” ICI, and extend their therapeutic landscape.

It is believed that one of the major reasons for which the success of ICI is limited is that therapy is applied on diseases that either lack infiltration of effector CD8^+^ T cells or include massive accumulation of T_regs_ that suppress their activities [26,31,32,33,34,35,36]. These tumors are known as “cold” tumors [49,50,51]. Turning “cold tumors” into “hot tumors” by enhancing the activity of tumor-specific infiltrating effector T cells may extend the relative number of responders to ICI [49,50,51,52]. Likewise, in tumors enriched with T_regs_, it is likely that blocking their activity or depleting these cells from the TME would turn cold tumors into hot.

The current review focuses on two key chemokine pathways, with relevance to cancer immunotherapy. The first refers to the CXCR3-CXCL9/CXCL10 pathway [53,54,55]. The other includes a selective targeting of CCR8 and interference in the CCR8-CCL1 pathway as this pathway drives the activity of T_regs_ [56]. Both approaches are complementary and are the major focus of the current review.

## 2. Key Chemokine-Chemokine Receptor Interactions That Support Tumor Growth

### 2.1. The CXCR4-CXCL12 Pathway

CXCR4 is a chemokine receptor with a single ligand, CXCL12 (stromal cell-derived factor-alpha, SDF1-a), that also binds CXCR7 [57]. The CXCR4-CXCL12 pathway is involved in many biological features associated with cell migration and homeostasis, among them homing of bone marrow stem cells, activation of adhesion receptors such as the alpha-4 beta-1 integrin VLA-4, neutrophile homeostasis, and others (for general review see [58]). The major interest in this chemokine as a potential target for cancer therapy flows from studies that mostly investigated the direct interaction between CXCL12 and CXCR4 on cancer cells. CXCR4 is largely expressed on many human tumors, that also express CXCL12, among them: lung, breast, cancers of the brain, colon, and colorectal cancer, pancreas, prostate, ovarian, leukemia, and melanomas [5,59,60,61,62,63]. In many of these cancers, this interaction is essential to support tumor survival, growth, and metastasis formation [5,59,60,61,62,63] (for a very recent review see [9]). Aside from that CXCR4 is largely expressed by epithelial cells and mediates epithelial cell migration via the activation of Rac1, matrix metalloproteinases MMP-14 and MMP-2, and increases the motility of cancer cells through the up-regulation of NF-κB and ERK-dependent pathway [64,65]. Long ago we observed that CXCL12 upregulates IL-10 production by macrophages, and T regulatory-1 cells (Tr1) and by so doing restrains the autoimmunity [66], and may suppress anti-tumor immune reactivity. More recently Chen et al. used combined therapy that also included a small molecule inhibitor of CXCR4 (AMD3100) to limit IL-10 production within the TME during the hepatocellular carcinoma (HCC) [67]. Thus far, CXCR4 has been a major target for the therapy of different cancers, either by blocking antibodies or small molecules, among them BKT140, bicyclam AMD070, AMD3100, AMD11070, MSX-122, GSK812397, KRH-3955, and several small modified peptides (reviewed in [9]).

### 2.2. The CCR2 Pathway

CCR2 is a chemokine receptor to numerous ligands including CCL2, CCL7, CCL8, and CCL13. Of them, CCL2 is its major ligand and exclusively binds CCR2 [68]. The role of CCR2-CCL2 interactions in directing the migration of CCR2^+^ monocytic cells has been mostly studied for two types of diseases: inflammatory autoimmune diseases, and cancer. Targeting this interaction restrained inflammatory autoimmunity by limiting the accumulation of inflammatory macrophages at autoimmune sites [69,70,71,72,73]. As for cancer diseases, the CCR2-CCL2 is likely to be involved by two different mechanisms: directing the recruitment and accumulation of TAMS and monocytic myeloid-derived suppressor cells (monocytic MDSC) at the tumor site to support tumor growth and suppress the anti-tumor immune response, and direct effect on tumor growth. Thus, targeted neutralization of CCR2 or blockade of CCL2 inhibited the recruitment of TAMS and monocytic MDSC at tumor sites, angiogenesis, cancer development, and metastasis in various cancer models, among them breast cancer, lung cancer, ovarian cancer, and others [74,75,76,77,78,79,80,81,82,83]. In contrast to this Bonapace et al. reported that neutralizing of CCL2 may aggravate breast cancer in an experimental model [83]. The association between low and high expression of CCL2 or CCR2 and cancer prognosis has been studied for several human diseases. Most of them show a clear link between high CCL2 expression and bad cancer prognosis, or low CCL2 expression and good prognosis, or polymorphism of CCR2 and prognosis of cancer diseases [84,85,86,87,88,89,90,91,92,93,94]. Finally, macrophage depletion increases the therapeutic efficacy of anti CTLA4 and anti-PD1 antibodies in mouse pancreatic cancer models [95]. As targeting the accumulation of TAMS is a key mechanism by which blockade the CCR2-CCL2 acts, it is possible that blocking this pathway, in combination with ICI would be beneficial compared to monotherapies.

### 2.3. The CCR5 Pathway

CCR5 is a chemokine receptor with three ligands: CCL3, CCL4, and CCL5. Of them, only CCL4 exclusively binds CCR5, whereas CCL3 also binds CCR1, and CCL5 also binds CCR1 and CCR3 [68]. CCR5 and its ligands, particularly CCL5 have been thought to support tumor growth either directly, by driving the migration of tumor cells to tumor sites, and mostly by recruiting MDSC and dendritic cells to the tumor site [96,97,98,99,100]. Many studies showed a clear link between CCR5 and CCR5 ligands polymorphism or differential gene signature and several cancer diseases, among them: prostate cancer, breast cancer, glioblastoma, myeloid leukemia, pancreatic adenocarcinoma, Non-Small Cell Lung Cancer (NSCLC), metastatic melanoma, metastatic colorectal cancer and others [98,99,101,102,103,104,105,106,107,108,109,110,111,112,113,114,115,116,117,118,119,120,121,122,123,124,125,126,127,128]. Following Balistreri et al. [129] that observed a very low prevalence of prostate cancer in men with CCR5 delta 32 mutations, which also confers HIV resistance, we have used mice lacking CCR5 to uncover the contribution of CCR5 to the cancer resistance and found that in these mice accumulation of MDSC at the tumor site is inadequate [130]. We further examined the mechanism by which CCR5 is involved in MDSC recruitment at the tumor site. MDSC are comprised of monocytic MDSC and polymorphonuclear MDSC (PMN-MDSC) [131,132]. The relative number of PMN-MDSC at the tumor site is much higher, and they are thought to be key drivers of immune regulation of anti-cancer immunity that limits tumor growth [131,132]. Accumulation at the tumor site of both types of MDSC is a multi-step event [133]. It starts with crosstalk between the tumor site and the hematopoietic stem and progenitor cells (HSPCs) at the bone marrow (BM) and secondary lymphatic organs, resulting in rapid myelopoiesis followed by mobilization to the blood. Although myelopoiesis is coordinated by several cytokines and transcription factors, mobilization is selectively directed by chemokine receptors and may differ between M-MDSC and PMN-MDSC. These myeloid cells may then undergo further expansion at these secondary lymphatic organs and then home to the tumor site (recently reviewed by us [133]). The mobilization of monocytic MDSC from the BM and secondary lymphatic organs to the blood, and later to the tumor site, is largely controlled by the CCR2-CCL2 interaction [81,134,135]. We have shown that the reciprocal mechanism that directs the mobilization of PMN-MDSC is CCR5-dependent, thus mice lacking CCR5 display a high state of resistance to cancer diseases that could be overridden by adoptive transfer of MDSC from WT mice [130]. Later, together with Viktor Umansky and his group, we extended the relevance of this observation to melanoma [136]. Recently, Yang et al. reported that blockade of CCR5 markedly suppressed the accumulation of myeloid cells at the tumor site in a mouse model of gastric cancer. This enhanced the anti-PD-1 efficacy in this model, implicating possible usage of anti CCR5 in combined ICI therapies [137]. Finally, clinical trials in which CCR5 is blocked in patients with colon cancer and breast cancer using a CCR5 small-molecule blocker are now being conducted [96,138].

### 2.4. The CXCR1/2 -CXCL8 Pathway

CXCR1 is a chemokine receptor with two ligands, CXCL6 and CXCL8. CXCL8 also binds to CXCR2 [68]. The role of CXCR1/2 -CXCL8 interaction with cancer has been studied from different perspectives including migration of neutrophils, TAMS, and CXCR1/2+ cancer cells to tumor sites [139,140] angiogenesis [141], and tumor stemness [142]. Altogether this makes the CXCR1/2 -CXCL8 pathway a target for cancer therapy [139,140,141,142,143,144,145,146,147]. Breast cancer is among the most relevant human diseases for the blockade of the CXCL8-CXCR1/2 interaction since CXCL8 is up-regulated in breast cancer patients is associated with poor prognosis [148,149], and CXCL8 drives tumor stemness and angiogenesis [146,150,151,152]. Along with this, Reparixin, a CXCR1 inhibitor, was effective in treating several NOD/SCID mice breast cancer models [146]. Currently, the extension of this study to human clinical trials is ongoing [153]. This therapy might be relevant to other cancers among them: gastric cancer [145,154,155], melanoma [156,157,158,159,160,161,162,163,164,165], and others.

## 3. Key Chemokine-Chemokine Receptor Interactions That Limit Tumor Growth

### 3.1. CXCR3 and Its Ligands

CXCR3 is a chemokine receptor that is primarily expressed on CD4^+^ and CD8^+^ T cells, and to some extent by other cells, among them, macrophages [166], NK cells [167,168,169], and epithelial cells [170]. Within the CD4^+^ subset, CXCR3 is most abundant on effector T cells, but notably, it is also expressed by FOXp3+ regulatory T cells (T_regs_) In humans, three isoforms were identified: CXCR3A that is reciprocal to the mouse CXCR3 and binds CXCL9, CXCL10, and CXCL11, CXCR3-B that binds CXCL9, CXCL10, CXCL11 as well as an additional ligand CXCL4, and CXCR3-alt that only binds CXCL11 [171]. The CXCR3 ligands share limited sequence homology. Yet, in their structural homology, there is more similarity between CXCR3 ligands as compared to other non-ELR chemokines. Additionally, all three chemokines are inducible by IFN-γ [172]. Together this makes them a well-characterized subfamily of the non-ELR chemokines. CXCL11 is believed to be the dominant CXCR3 agonist, as it is more potent than CXCL10 or CXCL9 as a chemoattractant and in stimulating calcium flux and receptor desensitization [173,174].

### 3.2. Biased Signaling via CXCR3 Drives Activity of CD4^+^ and CD8^+^ Subsets in the Context of Cancer and Autoimmunity

Biased signaling (also referred to as functional selectivity) means that a single receptor may transmit different signal transduction pathways, usually in response to different ligands. It was first associated with G-protein coupled receptors long ago by Luttrell et al. [175]. Very recently the molecular basis of biased signaling via G-protein coupled receptors has been uncovered [176]. As for chemokine receptors, several studies indicate the numerous chemokine receptors among them CCR7, CCR5, and CXCR3 induce biased signaling [177,178,179,180,181,182,183,184,185,186,187,188]. Seven years ago, we have shown that CXCR3 not only induces biased signaling induced by its different ligands, but also that this differential signaling shapes the biological function of effector T cells, and Tr1 cells [189,190]. We observed that CXCL11 differs in its binding site to CXCR3 from CXCL10 and CXCL9 also differs in signaling cascades and the biological functions resulting from this signaling. While CXCL9/CXCL10 potentiates effector T cells, CXCL11 induces, via a different signaling cascade, FOXp3-negative IL10-^high^ CD4^+^ T regulatory-1 cells (Tr1) [190,191]. This study was an outcome of early studies started almost 20 years ago, using targeted DNA vaccines technology and autoantibodies generated using this approach, we reported that CXCL10 and perhaps CXCL9, but not CXCL11 are associated with the induction of IFNg^high^ CD4^+^ effector Th1 cells, and therefore targeted neutralization of CXCL10 may restrain T cell-mediated autoimmunity [192,193]. More recently, Groom et al. showed that CXCL9 and CXCL10 are strongly related to the Th1-biased response that is a crucial part of the effector anti-tumor responses [194]. Independently, we observed biased signaling of CXCL9/CXCL10 and CXCL11 in CXCR3^+^ effector T cells that include potentiation of IFNg^high^ effector cells via CXCL9 and CXCL10, and potentiation of FOXp3-negative IL10^high^ in T-regulatory 1 (Tr1) cells via CXCL11 [190]. This motivated us to develop a fusion protein that includes murine IgG1 Fc linked to CXCL10 (CXCL10-Ig or CXCL10-Fc) for cancer therapy, and CXCL11 fused to the same construct for therapy of autoimmunity (CXCL11-Ig or CXCL11-Fc) [190]. While CXCL11-Ig based therapy inhibited inflammatory autoimmunity within the central nervous system [190], administration of CXCL10-Ig suppressed cancer [55].

### 3.3. CXCL10 and CXCL9 in Cancer Therapy: Could Systemic Administration of These Chemokines Induce Effector T Cells That Then Migrate to the Tumor Site to Limit Cancer Development?

Several studies, including our [55], have suggested three potential pathways for CXCL10 and CXCL9 in cancer diseases: 1. The immune-related pathway, 2. Direct suppressive effect on tumor growth, and 3. Inducing the ability of epithelial cells within the tumor microenvironment (TEM) to support tumor growth (Figure 1). Of these mechanisms we believe that the most important one is the immune-related mechanism because of its potential relevance for cancer immunotherapies, either as CXCL9 or CXCL10 based monotherapies or in combination with well-established immune checkpoint inhibitors (ICI) including anti-CTLA-4 mAb, anti-PD-1 mAb, and anti-PD-L1 mAb [21,22,23,25,26,28,29,33,37,195] to possibly increase the efficacy of response to ICI.

For immunotherapy, it is believed that the key target cells for CXCL10/CXCL9 based therapies are effector T cells, mostly CD8^+^ effector-cytotoxic T cells [53,167,174,196,197,198,199]. The traditional concept is that CXCL10/CXCL9 largely produced at the tumor site is associated with directing the migration of CXCR3^+^ effector CD4^+^ and mostly effector -cytotoxic CD8^+^ T cells, and CXCR3^+^ NK cells to the tumor site [167,174,197,198,199,200,201,202]. Recently it has also been shown that at the tumor site TGF b suppresses CXCR3 expression by CD8^+^ T cells thus enabling tumors to escape CXCL10 induced recruitment to the tumor site [196]. Along with this, early studies indicated that either overexpression of CXCL9 by cancer cell lines [203], or direct injection of CXCL10 to the tumor site [204], or targeted gene therapy of CXCL10 [205], may limit cancer development. The importance of CXCR3-ligands in the recruitment of tumor-infiltrating lymphocytes (TILs) to the TME had also been demonstrated in several human cancers in which CXCR3^+^ TILs were abundant and high levels of CXCL9 and CXCL10 were secreted by stromal cells [206,207]. This has also been associated with a better prognosis and enhanced survival [206,207]. The rational translational outcome of these studies is that CXCL9/CXCL10 based therapies would be effective if CXCL9 or CXCL10 would be administered or overexpressed intratumorally. Based on this statement it could well be that peripheral administration or systemic enhancement in the periphery of CXCL9 or CXCL10 could potentially lead to an opposing effect. That is, directing the recruitment of CXCR3^high^ effector T cells away from the tumor site.

Our collaborative study with Israel Vlodavsky and his team was among the first studies showing that systemic administration of CXCL10-Ig restrains cancer diseases [55]. Aside from our study that used peripheral administration of CXCL10-Ig for cancer immunotherapy [55]. Two key manuscripts that were published in very leading journals, showed that systemic increase in CXCL10, either via epigenetic approach or via targeting the exopeptidase Dipeptidyl-peptidase 4 (DPP4) that induces post-translational modifications of CXCL10 that targets it activity, resulted in increased expression of CXCL10 systemically while limiting cancer development and growth [53,54]. Barreira da Silva et al. used in their study very similar models to those used by us (C57BL/6 mice engrafted with B16 melanoma line, or C57BL/6 mice engrafted MC38 colon cancer line) [54]. Our current working hypothesis is that peripheral administration of CXCL10-Ig or CXCL9-Ig would induce the activity of CXCR3^+^ effector CD4^+^ T cells, effector CD8^+^ T cells, and NK cells that are then recruited to at the tumor sites to limit cancer. At this point, we do not exclude the possibility that following peripheral administration some of the injected CXCL10-Ig or CXCL9-Ig would enter the tumor site, and therefore may have an additional contribution to CXCR3^+^ cell recruitment to the TEM [208,209].

As both CXCL10 and CXL9 exclusively bind CXCR3 and are thought to attract, and possibly potentiate effector T cells, it is an open-end question whether except for redundancy they do differ in some biological functions, particularly functions that are related to effector T cell potentiation and combined immunotherapies? Chow et al. recently reported that CXCR3KO mice respond poorly to anti-PD-1 therapy. Their study suggests that the CXCL9-CXCR3 interaction is critical for successful anti-PD-1 immunotherapy [198]. Finally, Peng et al. reported that PD-1 blockade enhances T-cell migration to tumors by elevating CXCL10 and possibly CXCL9 and thereby IFN-γ producing T cells [210].

### 3.4. Selections of Target Human Cancers for CXCL10/CXCL9 Immunotherapy

Among the parameters that may assist in selecting target disease for CXCL10 or CXCL9 based therapies are results of pre-clinical trials in experimental models of various cancer diseases, and selection based on human data from cancer patients regarding the association of CXCL10/CXCL9 expression at the tumor site, or sera levels of these chemokines, and cancer prognosis. Diseases in which high expression of CXCL9 or CXCL10 correlated with a good prognosis, and low expression with a poor prognosis are the best potential candidates for CXCL10 or CXCL9 based therapy. Moreover, if the sera level of CXCL9/CXCL10 may predict success in CXCL9/CXCL10 based therapy it could be used for future precision medicine.

Regarding animal preclinical studies, as CXCL9 and CXCL10 mostly affect anti-cancer immunity the immunocompetent models are favorable. Of these models, there are currently available transgenic models in which cancer is developed “spontaneously” such as TRAMP mice for prostate cancer [211], or *ret* transgenic mice for melanoma [212]. Cancer pre-line (melanoma) or clones (prostate cancer) are also available and used for models of cancer engraftment. More advanced models, and for chemokines that may also affect tumor growth the patient-derived xenotransplantation models (PDX) are thought to best predict success in therapy [213,214,215,216,217], though usually initial experiments are regularly elaborated in immunocompetent mice.

As for the association of CXCL9 and CXCL10 with cancer prognosis (Table 1): more than ten years ago Jiang et al. found a correlation between low transcription of CXCL10 shows poor prognosis in stages II and III colorectal cancer (CRC) [218]. The study examined snap-frozen CRC tissues by RT-PCR [218]. Later Li et al. showed, also by detecting mRNA levels, that in patients with rectal cancer that are CXCL10^high^ a better response to chemoradiotherapy could be recorded, suggesting a synergistic beneficial effect of both [219]. Another cancer disease in which high levels of CXCL10 were associated with a good prognosis is epithelial ovarian carcinoma (HGSOC). In patients with this disease high levels of a CXCL10 antagonist could be associated with a poor prognosis, [220]. An additional disease with the relevance of CXCL10 high levels and prognosis that has been recorded is Osteosarcoma (OS). Flores et al. showed better survival in patients with high levels of CXCL10 [221]. Lastly, Zhang et al. showed that in hepatocellular carcinoma (HCC) high levels of CXCL10 are associated with better prognostic and overall survival [222]. These diseases are potential candidates for CXCL10 based therapy. What about CXCL9 and cancer prognosis? Thus far most of the studies focused on the role of CXCL10 in cancer diseases, particularly associated with cancer prognosis. Yet few studies elaborated on this subject regarding CXCL9. Patients with ovarian carcinoma showed that high levels of CXCL9 are associated with effector CD8^+^ T cell recruitment and good prognosis [223]. ER-Negative Breast Cancer patients also showed a good prognosis associated with immune cells infiltration in suggesting CXCL9 as a potential biomarker for the prognosis of this disease [224]. Another recent study also reported that in breast cancer high expression of CXCL9 and CXCL10 is associated with a good prognosis [225].

Are there diseases in which high CXCL10/CXCL9 could not be associated with a good prognosis and low CXCL10/CXCL9 with a poor prognosis? Few publications challenge the concept of high CXCL10 or CXCL9 and a good prognosis. This included pancreatic cancer, and myeloma, and Breast Cancer Registry [226,227,228]. One of the possible explanations for such discrepancy is that perhaps even though the direct effect of CXCL10 and CXCL9 on effector T cells, and possibly endothelial cells is similar, their direct effect on tumor cells may vary. It could well be that this subject should be further elaborated using PDX mice.

### 3.5. Why Do Cancer Cells Also Produce Chemokines That May Limit Tumor Growth?

Chemokines and their receptors hold different biological functions, among them directing the immune system to generate effective responses against bacteria and viruses. Among these chemokines, the CXCR3 ligands CXCL9 and CXCL10 are of major interest as they direct targeted migration of effector CD4^+^ and CD8^+^ T cells and promote their activities along with viral infections [229,230,231,232,233,234]. CXCL10 is largely produced by monocytic cells, endothelial cells, and fibroblasts whereas CXCL9 is also largely produced by CD103^+^ dendritic cells within the TME [198]. Importantly, CXCL10 and CXCL9 are highly expressed by human cancer cells, and this expression is correlated with a good prognosis [218,219,220,221,222,223,224,225].

From a deterministic viewpoint of cancer evolution, cancer cells developed means of using chemokines and their receptors to support tumor growth, metastatic spread, and escape from immune eradication. Key examples are the CCR2-CCL2 pathway, CXCR4-CXCL12 pathway, CCR2-CCR3 ligands pathway. Many cancer cells express these receptors and produce their ligands that support tumor growth, metastatic spread, and recruitment of bone marrow-derived cells that are recruited at the TME to support tumor growth and suppress anti-tumor immunity. It is somehow puzzling that cancer cells also produce CXCL9 and CXCL10 and that this correlates with a better prognosis, in part due to the induction of anti-tumor CD4^+^ and CD8^+^ T cells [197,218,219,222,235]. One possibility is that they produce these chemokines primarily to attract tumor cells to metastatic sites, as has been recently suggested for melanoma metastasis into the brain [236]. These chemokines are then being neutralized at the tumor site by the post-transcriptional modifications (PTM) mechanisms [237].

## 4. Regulatory T Cells in Cancer Diseases, and Chemokine Receptor-Based Selective Depletion of These Cells for Cancer Immunotherapy

Maintenance of immunological self-tolerance by suppressing self-reacting T cells, as well as restraining the activities of effector T cells in response to infectious stimuli, thus, limiting chronic inflammatory conditions, is largely regulated by CD4^+^ regulatory T cells; [238,239]. These cells fall into two major subsets: those that express the transcription factor forkhead box P3 (FOXp3), also known as regulatory T cells (T_regs_), and those that are FOXp3-negative but produce high levels of IL-10, also known as T regulatory -1 cells (Tr1) [238,239,240]. Those that are FOXp3+ commonly do not express the IL-7 alpha chain CD127, which is essential for IL-7 signaling required for converting T cells into memory cells [241,242,243]. These cells are of major interest for their key role in regulating cancer disease, mostly in suppressing the anti-cancer reactivity of effector T cells [244]. There are three major approaches for inhibiting T_regs_ and their ability to limit anticancer effector T cells: 1. Blocking the migration and accumulation of T_regs_ at the tumor site. 2. Inhibiting their suppressive activities within the tumor site and 3. Depletion of T_regs_ within the tumor site. Of these approaches, depleting T_regs_ is likely to be the most dramatic and possibly effective way. Yet systemic depletion of T_regs_ may result in major impairment of immune regulation. For example, a loss-of-function mutation in the gene encoding FOXp3 leads to a very severe autoimmune syndrome in humans named immune deficiency poly-endocrinopathy enteropathy X-linked (IPEX) syndrome [245].

Chemokines and chemokine receptors are thought to be involved in the selective migration of T_regs_ to the tumor site, and also in their potentiation within this site. T_regs_ express several chemokine receptors among them: CCR8, CCR4, CXCR3, CCR2, CCR6, and CCR5 [246]. Among these receptors, the CCR4-CCL22/CCL17 and the CCR8-CCL1 axis have been of major interest for both selective migrations of T_regs_ to tumor sites and their potentiation there. Moreover, their selective accumulation within the tumor site may suggest that selective depletion of CCR4^+^ or CCR8^+^ T_regs_ may enhance anti-cancer immunity while having a very limited effect on T_regs_ in the periphery. This subject is further discussed below.

### 4.1. CCR4^+^ T_regs_

CCR4 is a chemokine receptor with two ligands CCL22 and CCL17. Both ligands but mostly CCL22 are largely involved in directing the recruitment and induction of suppressive function of T_regs_ at the tumor site [247,248,249,250,251,252,253,254,255,256]. This includes breast cancer, cervical cancer, glioblastoma, squamous cell carcinoma (SCC) colorectal cancer (CRC), and Pancreatic ductal adenocarcinoma (PDAC) [247,248,249,250,251,252,253,254,255,256]. Aside from T_regs_, CCR4 is present in other leukocytes, among them CD4^+^ Th2 cells, NK cells, and macrophages [255,257,258,259]. It is also abundant on cancer cells, among them breast cancer [252]. Olkhanud et al. used a highly metastatic breast cancer (4T1) model in which CCR4 is largely expressed on cancer cells and T_regs_, and demonstrated the pivotal role of CCR4 in recruiting and inducing NK cells and T_regs_ to limit tumor development and metastatic spread [256]. This does not exclude the possibility that targeting CCR4 would be more effective in several cancer diseases in which cancer cells are also CCR4^+^, among them breast cancer. In human cancers, major target diseases are several solid tumors, B-cell lymphomas, T-cells lymphomas, and leukemia in which not only CCR4 is highly expressed within the tumor microenvironment by T_regs_, NK cells, and tumor cells, but mostly in those that poor prognosis has been associated with high expression of CCR4 on these cells [251,252,260,261,262,263]. Currently, there are two small chemical class II antagonists produced by Astra-Zeneca that block T_regs_ recruitment (AZD-2098, Marketed, and AZD-1678 in preclinical studies), a small chemical class II antagonist that blocks the interaction between CCL22 and CCR4 (FLX-475 produced by FLx-Bio) in phase 1/2 clinical trials as monotherapy or in combination with anti-PD-1 (Merck), a humanized mAb (KW-0761) capable of inducing ADCC to CCR4^+^ cells in phase 1a monotherapy for solid tumors, in combination with anti-PD-1 (Merck) for B cell lymphoma (phase 1/2), in combination with anti-PD-L1 or anti-CTLA-4 (Astra-Zeneca) in Phase 1b for solid tumors, and combination with anti-PD-1 (BMS) for solid tumors (very recently reviewed in [246]).

### 4.2. CCR8^+^ T_regs_

CCR8 is a chemokine receptor mostly, but not exclusively, expressed by FOXp3+ T_regs_ [257,264,265,266,267,268]. Human CCR8 has four known ligands: CCL1, CCL8, CCL16, and CCL18 [269], whereas in murine only 3 of them are expressed: CCL1, CCL8, and CCL16 [270,271,272]. In both humans and mice, CCR8 is the only known receptor for CCL1 [265], whereas the other CCR8 ligands bind several chemokine receptors, as well as decoy receptors [270,271,272]. Four years ago, we identified CCR8^+^ T_regs_ as master drivers of the immune regulation [56]. In this study, we observed that the relative number of CCR8^+^ T_regs_ that is very low in the periphery increases along with the development of experimental autoimmune encephalomyelitis (EAE), a T cell-mediated autoimmune disease of the central nervous system (CNS). This study also observed that within the CNS CCR8^+^ T_regs_ are potentiated by CCL1, possibly in an autocrine manner, which makes them “driver” regulatory cells that restrain the progression of the disease [56]. Independently, Plitas et al. showed that in several human tumors, particularly “cold tumors” such as breast cancer, that these cells are highly dominant [273]. Along with this, recently it has been reported that anti-CCR8 mAb could be used to limit cancer growth in several cancer models [274,275,276]. One of the major reasons for which the success of ICI is limited is that therapy is applied on diseases that are designated as “cold tumors” that either lack infiltration of effector CD8^+^ T cells, or include massive accumulation of T_regs_ that suppress their activities [49,50,277,278,279,280]. Anti CCR8 mAb, mostly depleting antibodies, are currently under preclinical development by several lead companies.

## 5. Could Chemokines Be Used to Turn Cold Tumors into Hot?

Tumors described as “hot” are those that show signs of inflammation, particularly massive infiltration and enrichment with effector T cells, the vast majority of the CD8^+^ T cells. The tumor cells have undergone many mutations that create neoantigens recognized by these T cells. For this reason, hot tumors typically respond well to immunotherapy treatment using checkpoint inhibitors [281,282,283,284,285,286,287,288,289,290]. It is believed that tumors that lack effector T cells (i.e., cold tumors) will fail in responding to ICI. The typical cancers that are considered as hot tumors and respond well to ICI are melanoma, bladder, kidney, head and neck, and non-small cell lung cancer—and further limiting the efficacy of immunotherapies is the fact that not every hot tumor in every patient will be responsive to such treatments [28,291,292,293,294,295,296]. For example, within melanoma patients, those that are specifically designated as patients bearing “hot” tumors preferentially respond to ICI as opposed to those with “cold tumors” [52]. Many cancers among them breast cancers, ovarian cancer, prostate cancer, pancreatic cancer, and glioblastomas are typically cold tumors. In many of these tumors, the microenvironment contains myeloid-derived suppressor cells (MDSC) and T_regs_. Which are known to dampen the immune response and inhibit T cells trying to move into the tumor [297]. One way of trying to override these cold tumors is by combining progressive immunotherapy treatments with traditional therapies such as radiation and chemotherapy. An alternative option, suggested herein is the “chemokine-based approach” of using CXCL10 and CXCL9 based therapies to enhance the activity of effector T cells that would then be recruited to the tumor site, possibly combined with selective elimination (depletion) of T_regs_ via anti CCR4 or CCR8 mAbs.

The concept of using Fc-based stabilized chemokines for therapy of either cancer or inflammatory autoimmunity is relatively new, and to our best knowledge has not been explored in human clinical trials yet. We first applied this technology thirteen years ago when generating stabilized CXCL12-Ig for therapy of autoimmunity within the CNS [66]. The basic concept has been that CXCL12 would enhance the development of L-10 producing Tr1-like cells and M2 macrophages and by so doing restrains autoimmunity within the CNS [66]. The idea of promoting it to clinical trials of different inflammatory autoimmune diseases has been omitted due to the pleiotropic function of this chemokine [58]. Subsequently, we proposed to use the Fc stabilized form of the CCR8-ligand CCL1 for induction of T_regs_ in the context of inflammatory autoimmunity [56]. It is yet to be studied if CCR8 is upregulated on T_regs_ in human inflammatory autoimmunity, as it doses in cancer diseases [273]. Stabilized CXCL9 and CXCL10 are likely to be good candidates for immunotherapy of cancer diseases. The two key open-end questions are which human cancer to treat, and which of them CXCL9 would be favored over CXCL10, and when CXCL10 would be used as a lead molecule over CXCL9? And if to use each of them as monotherapy or in combination with other ICI? Future preclinical studies are needed to explore these questions.

## 6. Conclusions

The most successful approach for cancer therapy over the last decade has been the use of monoclonal antibodies (mAb) to immune checkpoint inhibitors (ICI). Yet for many cancers, ICI shows limited success. Several lines of evidence imply that this limited success is mostly associated with attempts to treat patients with “cold tumors” that either lack effector T cells, or in which these cells are markedly suppressed by T_regs_. The current review focuses on two complementary approaches to possibly overcome this obstacle. The first includes the administration of two chemokines that potentiate the activity of effector CD4^+^ and CD8^+^ T cells and the other that selectively deplete CCR8^+^ regulatory T cells that are likely to be the dominant regulatory T cells within the TME of several human tumors, among them breast cancer. Future clinical trials will show if any of these approaches, or both could be used for cancer immunotherapy either as monotherapy or in combination with clasica ICI.

## Figures and Tables

**Figure 1 cancers-13-06317-f001:**
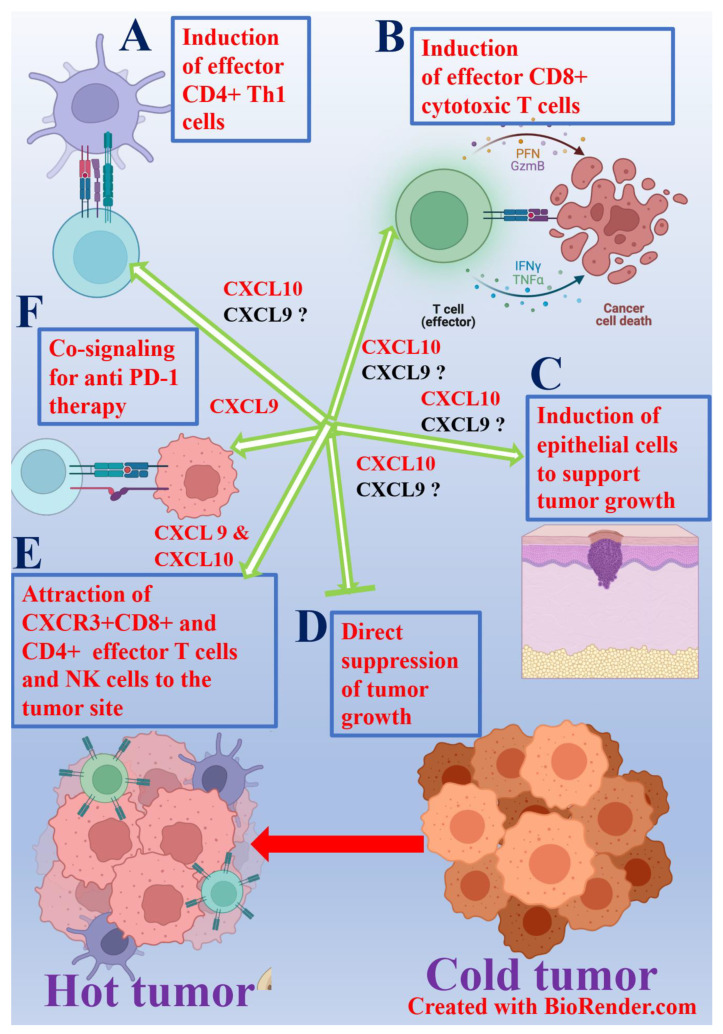
The role of CXCL9/CXCL10 in cancer diseases CXCL9/CXCL10 affect cancer diseases by either: inducing IFNγ^high^ CD4^+^ Th1 cells (**A**), cytotoxic CD8^+^ T cells (**B**), inducing growth factors via CXCR3^+^ epithelial cells (**C**), direct suppression of tumor growth (**D**), The attraction of CXCR3^+^ T cells and NK cells to the tumor site (**E**) and co-signaling with anti-PD-1 (only CXCL9) (**F**).

**Table 1 cancers-13-06317-t001:** Association of CXCL10/ CXCL9 with cancer prognosis in human.

Disease	Prgnostic Association	Reference
Colorectal cancer (CRC)	low transcription of CXCL10 and poor prognosis in stages II and III CRC examined by snap-frozen CRC tissues by RT-PCR.	[218]
Rectal Cancer	Patients that are CXCL10^high^ (RT PCR) display a better response to chemoradiotherapy, suggesting a synergistic beneficial effect of both	[219]
Epithelial ovarian carcinoma (HGSOC)	In patients with this disease high levels of a CXCL10 antagonist could be associated with a poor prognosis	[220]
Osteosarcoma (OS)	Better survival in patients with high levels of CXCL10 in circulating blood	[221]
Hepatocellular carcinoma (HCC)	High levels of CXCL10 in tumor tissues were associated with better prognostic and overall survival	[222]
Ovarian carcinoma	High levels of CXCL9 are associated with effector CD8^+^ T cell recruitment and good prognosis	[223]
ER-Negative Breast Cancer	Good prognosis associated with immune cells infiltration in suggesting CXCL9 as a potential biomarker for the prognosis of this disease	[224]
Breast cancer	High expression of CXCL9 and CXCL10 is associated with a good prognosis	[225]

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
