# Peer review of "Chemokines in the Landscape of Cancer Immunotherapy: How They and Their Receptors Can Be Used to Turn Cold Tumors into Hot Ones?"

_cancers, 2021, doi:10.3390/cancers13246317_

Round 1

Reviewer 1 Report

The strength of this article is well organized for readers to understand for issue of cancer immunotherapy in cold and hot tumor.
Theme of this review, to suggest a better therapeutic approach of cancer immunotherapy either as monotherapy or in combination with clasicaimmune checkpoint inhibitors. However, all authors should be cautious about “results were easy to anticipate”.
It's just my opinion, i think it would be good if your review could expand into the field of treatments for poor prognosis cancer like cancer stem cells.
In line 272, ‘CXCR3 and its ligands’. I think in a significant number of patients never response (resistance case) to this therapy or experience disease progression after an initial response (acquired resistance). It's just my opinion, if your next review could expand to fundamental challenge in the field of the identification biomarkers that predict response or resistance to these immune checkpoint inhibitors (ICI).

  1. In this study, author proposed two key chemokine pathways, with relevance to cancer immunotherapy.
  2. Only minor points to be considered: English language should be revised throughout the text.
  3. Line 67, The period overlapped. Please correct to ‘..’ > ‘.’
  4. Line 116, ‘((“ > ‘(‘
  5. Line 299, reference might be mistake. Please check it up again
  6. This paper is well organized for readers to understand. A main theme is so interesting too.

Author Response

I would like to thank reviewer #1 for reviewing my manuscript. I have corrected all minor corrections. In my next review on the field, I will surely add a major paragraph on expanding to fundamental challenges in the field of identifying biomarkers that predict response or resistance to these immune checkpoint inhibitors (ICI). 

thanks again for reviewing this manuscript

Reviewer 2 Report

In this paper by Karin, the author provides an extensive and in depth review about the role of chemokines and chemokine receptors in cancer progression and therapy.  This is a needed review since there at only a small number of broad reviews on this topic in the last five years.  Overall, the review is well written.  There are only a few minor comments that would enhance the readability of the review.

  • It would be good for the review to undergo a thorough edit. There are several minor grammatical errors that did distract from the overall, well-written review.
  • It is clear that the places where the author had directly contributed to the field received more focus. I would suggest that the author think about whether there are opportunities to provide similar depth in other sections with high clinical relevance.
  • It was interesting that the author chose to use the term we when the paper was written by a single author.
  • I would have like to seen more detail in section 5 of the review. The idea that chemokine therapy could be used to alter the immune landscape of tumors is intriguing.  A major point of this review should be to posit this hypothesis in depth.

Author Response

 I would like to thank reviewer #2 for reviewing the manuscript and for the constructive comments.   We have revised the manuscript as follows:

  1. I would suggest that the author thinks about whether there are opportunities to provide similar depth in other sections with high clinical relevance". I tried to focus the manuscript on two types of chemokine-chemokine receptors interactions, which we find a major interest, and that we believe are directly related to turning cold tumors into hot" I intend in the near future to try writing a wider review focusing on chemokines and their clinical relevance in cancer that will provide a wider depth on chemokines and cancer therapy.
  2. I mostly used "We" when referring to work done in my lab. In other parts, I tried to be polite. I am looking for a collaborator for writing together with an additional review. 
  3. Section 5 has been extended. 
  4. Again I would like to thank reviewer #2 for the constructive comments that helped me to improve the manuscript. 

Reviewer 3 Report

Nathan Karin discussed the role of chemokines and their potential to turn cold tumors into hot ones, additionally leading to improved cancer immunotherapy which can significantly increase response rates of cancer patients. The author provided a perspective on the usage of chemokines as a combinatorial therapeutic approach.

Overall, the review is nicely written with lots of useful information. However, this review needs to be reorganized a bit and the references need to be checked. I would suggest to change chapter 3 (Tregs) and 4 (chemokines) to keep the flow on chemokines (after chapter 2).

Fig1: The key message from Figure 1 is nice and the overall design is good. Please refine and redesign as it is not ready for publication in its current form (attention on upper and lower case, uniform arrows, no overlapping of arrows over text …)

page1, line20/21: Two sentences started with „This may …” – please rephrase

p2, l53: rethink BMDR – maybe use BM (bone marrow) derived cells

p2, l74: refine references of limited ICI success – more actual literature

p2, l85: Ref 23 not addressed to Tregs – refine

p3, l110: delete “mediates” once

p3, l124: delete “CCR2” once

p3, l134: Ref81 not associated with CCR2

p3, l140: delete “diseases” once

p3, l141-144: rephrase sentence

p4, l177-180: Ref134 Yang et al refers to gastric cancer – not CRC, change text

p4, l187: Ref [136, 137] – both in one bracket

p6, l275: Ref 206 – CXCR3 not found on DCs?

p6, l294: Ref 219 – wrong citation, not Lefkowitz

p6, l299: check REFs

p10, l424: would suggest to write breast cancer not Ontario Familial Breast Cancer Registry

Author Response

I would like to thank reviewer #3 for reviewing the manuscript and for the positive comments based on which I have revised the manuscript as follows:

  1. Chapters 3 and 4 have been changed so chapter 4 (chemokines) comes before chapter 3.
  2. Fig 1 was revised as suggested.
  3. All typos were revised.I would like to thank again reviewer # 3 for the comments.